Empirical study on visual attention characteristics of basketball players of different levels during free-throw shooting

Zhao Chunzhou
Li Sunnan hanceli99@126.com
Zhao Xuetong
College of P.E and Sports, Beijing Normal University , Beijing , China
Lu Frank
Electronic publication date: 2023 Dec 11
Publication date: 2023
Volume: 11
Electronic Location ID: e16607
Received 2023 May 23; Accepted 2023 Nov 15
Copyright: © 2023 Zhao et al.
Copyright year: 2023
Copyright holder: Zhao et al.
License: This is an open access article distributed under the terms of the Creative Commons Attribution License, which permits unrestricted use, distribution, reproduction and adaptation in any medium and for any purpose provided that it is properly attributed. For attribution, the original author(s), title, publication source (PeerJ) and either DOI or URL of the article must be cited.
License URL: https://creativecommons.org/licenses/by/4.0/

Keywords: Free throw, Eye movement, Gaze pattern, Information processing.

Funding: 2022 China Social Science Fund Project 22BTY055 The present study was supported by the ‘‘2022 China Social Science Fund Project” (Project No.: 22BTY055). The funders had no role in study design, data collection and analysis, decision to publish, or preparation of the manuscript.

==============================
Background

Visual attention is very important in basketball shooting, which is a prerequisite for forming good muscle proprioception and improving the shooting rate. The ability of visual instant searching information in the process of free throw plays an important role in the mobilization of free throw percentage. The aim of this study is to explore the fixation characteristics of athletes at different levels in the process of free throw shooting and to provide scientific basis for improving the free throw training of basketball players.

Methods

A total of 20 expert basketball players, 20 general basketball players and 20 novice basketball players participated in the experiment. Participants in the three groups wore eye tracker to make free throws, and analyzed the difference of visual attention among the three groups.

Results

The expert group had fewer number of fixations on the front, bottom, top-right, and top-right areas of the basket than the general and novice groups. The expert group also had smaller saccadic amplitudes than the other two groups. In terms of fixation duration, the expert group had longer fixation duration on the front and top areas of the basket than the other two groups, while in the top-right and bottom-right areas, the expert group had shorter fixation durations. The pupil dilation of the expert group was larger than that of the other two groups.

Conclusion

During the free throw process, the expert group had a clear attentional focus, concentrated fixation points, efficient information search, and precise processing. Profound basketball knowledge, proficient skills, and accumulated experience are the foundation of visual efficient search and precise processing during free throw attempts, and they are also the prerequisite for ensuring a high free throw shooting percentage.

Introduction

The eyes are the windows to the soul, providing valuable information for the study of human psychological processes. Eye-tracking technology is an effective, real-time method for exploring cognitive processing. This technology measures fixation, saccadic amplitude, and regression using visual input (Kathy, Sánchez & Carrol, 2020). In sports psychology research, the novice-expert paradigm is the most commonly used research paradigm for eye-movement technology analysis (Bai & Yan, 2018; Shen, Bai & Yan, 2008). Zhang et al. (2004) used the novice-expert paradigm to study the eye movement characteristics of basketball guard players and found that athletes with different skill levels had different fixation patterns and fixation allocation. Liao, Zhang & Ge (2011) used the novice-general-expert paradigm to study the eye movement of volleyball players. Zhang (2013a) found that having three groups of participants in the experiment leads to richer information and a more detailed quantitative analysis.

The free throw is a complex targeting skill that requires the integration of visual information gained through overt shifts of gaze with effector movements that aim the ball (Vickers, 1996). Shooting quality is mainly determined by the mechanics of shooting at the basket, the positions of the eyes and head, and concentration level (Netolitzchi et al., 2019). Jiang (2000) believes that visual attention is very important in basketball shooting and is a prerequisite for the formation of good muscle proprioception and improved shooting rates. In order to achieve and maintain a high free-throw shooting rate, it is important to maintain a high degree of attention, a stable fixation position, and to focus attention in the proper range when shooting. Wulf & Prinz (2001) identified attention as one of the most important factors in motor learning and performance. Concentration is a particularly important factor for self-paced (Singer, 2010) and closed-skill tasks (Loze, Collins & Holmes, 2001), such as accurate basketball free throws. Attention level determines information detection (Moeinirad et al., 2020), and focused attention leads to the best performance (Lewthwaite & Wulf, 2017; Wulf & Lewthwaite, 2016). Basketball players with a high free-throw percentage move their fixation appropriately to discover key visual information that may predict a shot (Mizuguchi, Honda & Kanosue, 2013). Elite NBA shooters, sometimes with their eyes blocked by the defense, are able to shoot the ball, mainly because they have fixed the specific target location in the early stage of the shooting action, and their brains have already memorized the target location when they shoot. This is due to long periods of intensive training and competition, optimizing the neural network so the task is controlled to improve the shooting percentage (Vickers et al., 2017). Current studies agree that visual attention is very important when shooting in basketball, including free throws. In learning or training, basketball players are taught to focus on aiming at the basket. However, in free throws, differences in the visual characteristics, information search, and information processing of players with different free-throw percentages are still unknown.

Due to equipment limitations, previous research has mainly used images and videos of basketball players, collecting data on the eye movement characteristics of basketball players through observation and then analyzing the data. These studies have provided an important theoretical basis and methodological support for further research. With the rapid development of technology, eye-tracking devices have greatly improved, especially with the advent of portable eye trackers, providing more convenient conditions for research in actual game scenarios. Building on previous research, this study adopts the novice-general-expert paradigm, hypothesizing that there are significant differences in the information search strategies and information processing levels of expert, general, and novice basketball players during free-throw shooting. Using an eye-tracking device to test the visual information of expert, general, and novice groups during free-throw shooting in a real-world context, this study compares differences in visual processing between expert, general, and novice basketball players. This study aims to summarize the visual patterns of experts during free-throw shooting, provide new methods for studying the visual characteristics of basketball players in real-world scenarios, and improve the validity of eye-tracking research in basketball. This study lays a theoretical foundation for building an eye-tracking model of high-level athletes during free-throw shooting and provides important scientific evidence for teaching, training, and improving basketball free-throw shooting accuracy.

Materials and Methods

Participants

The original plan was to select 60 participants to participate in this experiment. Considering that some people would quit during the experiment, we selected 21 participants in each group (one participant in each group as an alternative), and a total of 63 participants were selected. The participants were divided into three groups: expert, general, and novice. Twenty-one players from the Beijing Normal University women’s basketball team were selected as the expert group. These players were all China University Women’s League champions, and had free-throw shooting percentages of over 85%, an average training period of 10.2 years, and were an average age of 23.6 years. Twenty-one female students from the basketball specialty class of the Beijing Normal University School of Physical Education and Sports were selected as the general group with no sports skill level, a free-throw shooting percentage of 50–70% (these participants were selected based on the free-throw test), and were an average age of 20.7 years. Twenty-one novice participants were selected from female students at Beijing Normal University who had not received formal basketball training and had a free-throw shooting percentage below 40% (these participants were selected based on the free-throw test), and were an average age of 19.6 years. All participants had normal vision with a naked eye visual acuity of 1.0 or above. All participants volunteered to participate in the experiment. Before the experiment, the organizer of the experiment issued posters to recruit participants according to the requirements of the experiment, and the registration time was from August 31 to September 5, 2022. A total of 72 novice, 47 general and 23 expert participants were recruited. After registration, participants were selected according to the experiment requirements. The free-throw shooting percentage test will be held at the Beijing Normal University Gymnasium from September 9 to 12, 2022.

The experimental protocol was approved by the regional ethics committee of Beijing Normal University (No. 20221126). All of the participants provided written informed consent prior to the start of the experiment.

Apparatus

The experiment used the Tobii Glasses three portable eye tracker produced in Sweden, with a sampling rate of 100 Hz. This instrument does not cause any obstruction to the wearer’s field of vision and provides maximum freedom of head and body movement without compromising data quality to ensure the most natural and authentic behavior was captured (Gou & Li, 2023).

Experimental design

The experiment was designed as a between-subject design. The independent variable was the athletic ability level of the participants, which was divided into three levels (expert group, general group, and novice group). The dependent variables were eye movement indicators, including fixation duration, number of fixations, and pupil dilation.

Experimental set-up and procedure

Experimental set-up

This experiment was conducted on December 29, 2022, and was completed in the basketball hall of Beijing Normal University. The participants wore the eye-tracking device and stood at the free-throw line of the gymnasium basketball court, following the rules of the International Basketball Federation for free-throw shooting. Each participant shot three free throws in a row and data was recorded. There were no other distractions on the basketball court. In order to conduct the experiment smoothly, all participants were trained on issues related to the experiment before performing the experiment.

Experimental procedure

The entire experiment was conducted in a disturbance-free gymnasium and supervised by the experimenter. Prior to testing, the participants were randomly assigned a testing order through a drawing. Only one participant was allowed into the basketball court for testing at a time while the others waited in the rest area. Before starting the test, the participant stood at the free-throw line, and the experimenter explained the instructions and assisted the participant in wearing and calibrating the testing device. Then, the experimenter instructed the participant to dribble the ball 2–3 times before shooting for each round, while the operator and recorder of the device began to record. Each participant made three consecutive free throws, and left the court at the end of the test. Each participant was tested for about three minutes, and the entire experiment took about 200 min in total. For the data analysis, the data from each participant’s first free-throw shot were analyzed, with the data from the second and third free throws used as backups if the data from the participant’s first free throw were problematic.

Division of area of interest (AOI)

Area of interest (AOI) is the part of the visual field that the subject focuses on during information search. Based on the recommendations of basketball experts, scholars, and instrument engineers, this study divided AOI into seven regions based on the locations that the subjects fixated on during the free-throw shooting process (see Fig. 1).

Figure 1 AOI based on free-throw fixation position (revision).

Data analysis

The data results were expressed as total, mean, and standard deviations, and a variance analysis was performed using SPSS 26.0 (SPSS Inc., Chicago, IL, USA). If the effect was significant, post-hoc comparisons were performed using LSD. An alpha level of 0.05 was preselected for all of the statistical comparisons.

Results

Number of fixations in different AOI

Table 1 shows that as skill level increased, the average number of fixations in each area decreased. The analysis of variance found significant differences (p < 0.05) in the number of fixations of the three skill-level groups in the front (F(2,57) = 5.014, p = 0.010, η2 = 0.15) and along the bottom areas (F(2,57) = 5.314, p = 0.007, η2 = 0.16) of the basket and very significant differences (p < 0.01) in the number of fixations on the top right (F(2,57) = 40.336, p = 0.000, η2 = 0.59) and bottom right areas (F(2,57) = 29.801, p = 0.000, η2 = 0.51) of the basket. Further LSD tests revealed significant differences in the number of fixations between the expert and novice groups in the front area of the basket (p < 0.05), significant differences between all three groups of participants in the bottom area of the basket (p < 0.05), highly significant differences between all three groups of participants in the top right area of the basket (p < 0.01), and significant differences between the general and novice groups in the bottom right area of the basket (p < 0.05).

Table 1 Comparison of average number of fixations in each area of interest between groups of different skill levels (Revision).

AOI	Expert group	General group	Novice group			
M	SD	M	SD	M	SD	F	p value	
Front	1.30	0.47	1.50	0.51	1.80	0.52	5.014	0.010*	
Top	1.20	0.41	1.35	0.49	1.45	0.51	1.421	0.250	
Bottom	0.95	0.22	1.25	0.44	1.35	0.49	5.341	0.007*	
Top left	1.05	0.22	1.20	0.41	1.25	0.44	1.563	0.218	
Bottom left	0.90	0.30	0.95	0.22	1.00	0.00	1.036	0.361	
Top right	0.05	0.24	0.70	0.47	0.95	0.22	40.336	0.000**	
Bottom right	0.05	0.22	0.65	0.49	0.90	0.31	29.801	0.000**	
Notes:

* p < 0.05.

** p < 0.01.

Fixation duration

Fixation duration of different skill levels of participants during free throws.

Table 2 shows that the expert group had the longest average fixation duration in the front and top areas of the basket, the general group had the longest average fixation duration in the top left area of the basket, and the novice group had the longest average fixation duration in the bottom, bottom left, top right, and bottom right areas of the basket. Analysis of variance showed significant differences (p < 0.05) between the groups in the front (F(2,57) = 23.970, p=0.000, η2 = 0.45), top (F(2,57) = 37.345, p = 0.00, η2 = 0.58), top right (F(2,57) = 27.662, p = 0.000, η2 = 0.49), bottom right (F(2,57) = 25.476, p = 0.000, η2 = 0.49), and bottom right (F(2,57) = 4.023, p = 0.023, η2 = 0.11) parts of the basket. Further LSD tests showed highly significant differences between the expert and novice groups, and between the general and novice groups in the front area of the basket (p < 0.01), highly significant differences between the expert and novice groups, and between general and novice groups in the top area of the basket (p < 0.01), significant differences between the expert and general groups, and between the expert and novice groups in the bottom area of the basket (p < 0.05), highly significant differences between the expert and novice groups, expert and general groups, and general and novice groups in the top right area of the basket (p < 0.01), highly significant differences between the expert and novice groups, and between the expert and general groups in the bottom right area of the basket (p < 0.01), and a significant difference between the general and novice groups in the bottom right area of the basket (p < 0.05).

Table 2 Comparison of fixation duration of participants of different skill level (Revision).

AOI	Expert	General	Novice			
M	SD	M	SD	M	SD	F	p value	
Front	629	93	591	63	476	56	23.970	0.000**	
Top	563	117	543	54	365	50	37.345	0.000**	
Bottom	319	84	368	38	368	57	4.023	0.023*	
Top left	337	32	364	55	361	60	1.754	0.182	
Bottom left	225	85	230	65	250	50	0.740	0.482	
Top right	15	67	149	101	209	56	27.665	0.000**	
Bottom right	10	110	142	110	200	76	25.476	0.000**	
Notes:

* p < 0.05.

** p < 0.01.

AOI number of fixations and fixation duration distribution (Tables 3 and 4)

The sum and mean number of fixations and the total and mean fixation duration in the AOI were taken during free throws for all three groups of participants. Tables 3 and 4 show that the total number of fixations in the AOI of the expert group (100) was less than that of the general group (152) and the novice group (174). The fixations of the expert group were mainly distributed in the front (26%) and top (24%) areas of the basket, with almost no fixation in the top right and bottom right areas of the basket. In the novice group, the average fixation frequency in the top right and bottom right parts of the basket accounted for 11% and 10% of the total number of fixations, respectively, and 9% and 9% in the general group, respectively. This indicates that the fixation range of the novice group and the general group was larger than that of the expert group. The total AOI fixation duration in the general group was longer than that of the novice and expert groups. The total fixation duration of the expert group was the shortest, with fixation time mainly concentrated in the front and top (37%) areas of the basket. The general group had higher average fixation durations in the front (22%) and top (24%) areas of the basket than the novice group (21% and 17%, respectively). Both the novice and general groups had higher mean fixation durations in the top right and bottom right parts of the basket than the expert group, indicating that the expert group’s attention stability was better than that of the general group and the novice group.

Table 3 Total number of fixations, average number of fixations, and percentage of total fixations in each AOI of participants in the three groups (Revision).

AOI	Expert group	General group	Novice group	
Total	M	PCT (%)	Total	M	PCT	Total	M	PCT (%)	
Front	100	26	26	152	30	20	174	36	21	
Top	100	24	24	152	27	18	174	29	17	
Bottom	100	19	19	152	25	16	174	27	16	
Top left	100	11	11	152	24	16	174	25	14	
Bottom left	100	18	18	152	19	12	174	20	11	
Top right	100	1	1	152	14	9	174	19	11	
Bottom right	100	1	1	152	13	9	174	18	10	
Notes:

There are 20 participants in each group and 60 total participants in all three groups.

Total is the sum of the number of fixations in each AOI for each group of participants; mean refers to the average number of fixations of participants in each group in OAI; PCT is the percentage of the average number of fixations in each AOI of the total number of fixations in each group of participants.

Table 4 Total fixation duration, average fixation duration, and percentage of total fixation duration in each AOI of participants in the three groups (Revision).

AOI	Expert group	General group	Novice group	
Total	M	PCT (%)	Total	M	PCT	M	Total	PCT (%)	
Front	2,098	629	30	2,315	519	22	2,229	476	21	
Top	2,098	563	27	2,315	543	24	2,229	365	17	
Bottom	2,098	319	15	2,315	368	16	2,229	368	17	
Top left	2,098	337	16	2,315	364	16	2,229	361	16	
Bottom left	2,098	225	11	2,315	230	10	2,229	250	11	
Top right	2,098	15	1	2,315	149	6	2,229	209	9	
Bottom right	2,098	10	0	2,315	142	6	2,229	200	9	
Notes:

There are 20 participants in each group and 60 total participants in all three groups.

Total is the sum of the fixation duration in each AOI for each group of participants; mean refers to the average fixation duration of participants in each group in OAI; PCT is the percentage of the average fixation duration in each AOI of the total fixation duration in each group of participants.

Pupil dilation of participants of different skill levels during free throws (see Table 5)

Table 5 shows that during free throws, the expert group had the largest average pupil dilation and the novice group had the smallest. Analysis of variance revealed significant differences (p < 0.01) in average pupil dilation between the three groups (F(2,57) = 8.503, p = 0.001, η2 = 0.23). Further LSD tests revealed that the difference between the expert group and the novice group was very significant (P < 0.01), the difference between the expert group and the general group was significant (p < 0.05), and the difference between the novice group and the general group was not significant (p < 0.05).

Table 5 Comparison of pupil dilation of participants with different skill levels (Revision).

Group	M	SD	F	p value	
Expert	1,384.95	90.03	8.503	0.001**	
General	1,333.60	57.59			
Novice	1,300.00	38.65			
Notes:

*p < 0.05.

** p < 0.01.

Heatmap

The heat map uses different colors to show the participant’s attention to the target area, or to show the participant’s residence time in a certain area, which can display a large amount of eye movement data in the most intuitive form (Yan & Bai, 2018). The degree to which each group of participants paid attention to each area can be visually displayed using these heatmaps (Jin, 2020). The heatmaps (Fig. 2) show that during free throws, the focus points of the participants in the expert group were mainly concentrated at the front edge of the basket, and the concentration and stability of visual attention in the expert group were better than those in the general group and the novice group. The general group had better fixation stability and concentration than the novice group, and the novice group’s visual attention was scattered during free throws. These results indicate that visual attention stability has an important effect on free-throw percentage.

Figure 2 (A–C) Free throw heatmap of the expert player.

Discussion

Spatial dimension eye movement metrics

Visual fixation is the main way participants seek decision-making information and the number of fixations reflects the participant’s information search strategy. The total number of fixations of the expert group in each AOI was less than that of both the general group and the novice group, with the expert group’s fixation mainly concentrated in the front and top two regions of the basket, indicating the expert group’s information search strategy during free throws was more efficient than that of the general and novice groups. Visual search is a complex cognitive process, and eye fixation, indicating the brain acquiring external visual information, partly reflects visual search strategies for different cognitive tasks (Jin, 2020). Sun (2004) found that the ability to control attention range and direction is an important indicator of the psychological ability of basketball players that directly affects athletic performance. Expert athletes use practice-based knowledge to control their eye movement patterns to find and extract important information (Williams et al., 2002; Robin & Perer, 2007). Differences in the average number of fixations on the front, bottom, top-right, and bottom-right areas of the basket among participants of the three groups were significant, with the expert group having fewer fixations than the general and novice groups. This demonstrated the expert group’s targeted concentration on the areas that have a significant impact on free-throw shooting.

The expert group could quickly locate important information during free-throw shooting with concentrated fixation points and fewer fixations and regressions, demonstrating the efficient search pattern of expert athletes during free-throw shooting. These athletes knew which target area they should pay attention to and had a more mature information search mode. The novice and general groups had more fixations and dispersed fixation points. During free-throw shooting, these participants were unsure which areas provided the most relevant information, indicating a clear difference in their ability to search for, analyze, and synthesize information, and make correct decisions compared to the expert group. Williams, David & Williams (1993) also found that professional athletes have more appropriate and efficient visual search strategies than novice athletes. The results of the present study also found that the front area of the basket is an important area for visual search during free-throw shooting, with the expert group allocating fixations in this key area, demonstrating concentrated fixation trajectories and high efficiency in retrieving key information.

Lv & Shi (2020) suggest that specialized knowledge and competition experience are both important factors in the perceptual decision-making of athletes. To make quick and accurate decisions in constantly-changing sports scenarios, athletes need to quickly adjust their visual search strategies based on their competition experience and specialized knowledge in order to complete the given task faster and with better results. Therefore, technical knowledge of free throws, movement proficiency, and athletic experience significantly affect the information search strategy of players during free throws. Expert basketball players are proficient in their specialized techniques, have a high level of automation, and have formed stable visual search areas and characteristics during long-term training. These athletes are able to focus their attention and have a clear visual target and directionality. They can quickly lock onto specific visual areas within the target range during free-throw shooting, extract useful information, and respond appropriately after processing, reflecting an efficient visual search strategy. When expert participants are presented with information related to their expertise, this information activates memory traces, resulting in preferential attention allocation, giving these players stronger perceptual-cognitive advantages (He & Qi, 2022).

Expert basketball players have efficient search strategies during free throws because they are better able to encode and process specific information and predict the final landing point of the ball based on their rich cognitive information database (Williams et al., 1994). Coaches train basketball players in both the skills and psychological processes needed for free throws. In the free-throw training process, the player will visually screen information around the target and learn to efficiently process that information. This creates visual muscle memory for the player and, over time, the player will form a stable, automated visual scanning pattern for free throws (Yang, 2021; Tran & Silverberg, 2008). With tens of thousands of successful free-throw training sessions, the final, optimized visual pattern becomes a conditioned reflex. This visual pattern is also the key to a high-free throw success rate.

In this study, the expert group had fewer fixations in the bottom left, top right, and bottom right regions because they selectively focused on the key areas of interest that had the most impact on free-throw shooting accuracy, based on their trained information selection and validation processes. This demonstrates their directed and efficient visual strategy. Jin et al. (2020) explained that expert athletes develop a unique “knowledge block” of specialized sports knowledge through their extensive training and competition experiences. These athletes utilize their perceptual and cognitive advantages to extract the most valuable information from a vast amount of complex information, enabling them to quickly process information and make accurate judgments, resulting in a fewer number of fixations and a clear fixation trajectory. In contrast, the novice group lacked training and so had limited knowledge and skills related to free-throw shooting. This group had weak encoding and processing abilities, which made it difficult for them to accurately judge the landing point of the ball during free-throw shooting. Their visual search was unfocused and they were unable to quickly and effectively locate crucial visual information (Liu & Tang, 2022). As a result, they had a dispersed fixation pattern and searched for information across a larger area rather than concentrating their attention on key areas. The majority of their time was spent scanning for information, leaving no time for information processing.

Because the expert group had accumulated rich knowledge of free-throw shooting through years of scientific and systematic training, their information search was highly strategic, focusing more attention on key areas of interest. In contrast, the novice group had a scattered fixation pattern and complex fixation trajectories during visual information search in the free-throw shooting process. Differences in visual attention during free-throw shooting between different levels of athletes also stem from differences in task orientation and cognitive control, which are influenced by cognitive drivers (Carlisle & Woodman, 2011). The expert group and the general group had a clear visual attention target during free-throw shooting, indicating “goal-driven” visual attention. The novice group had no clear fixation target during free-throw shooting, blindly searching for visual information, indicating “stimulation-driven” visual attention (Riegler & Riegler, 2020). Mann et al. (2007) also found that in two different visual presentation modes (video presentation or still image presentation), expert athletes had a low gaze frequency and long fixation duration, indicating that regardless of the presentation mode, the visual search efficiency of expert athletes is higher than that of novice athletes.

Temporal eye movement metrics

Fixation duration reflects both the duration of fixation at each fixation point and the processing level of the information. Participants of different skill levels have different fixation durations on each area of interest. The longer the fixation duration, the longer the participant is gathering and processing information in a certain area of interest (Zhang et al., 2004). Paying too much attention to the details of the action during free-throw shooting can have a destructive effect on the smoothness of the action. When in a psychologically fluent state, attention can be focused on the target area (Zhang, 2013b). The total fixation duration of each area of interest in the expert group was not longer than that of the general and novice groups, but the distribution of fixation duration differed between the three groups. Significant differences in average fixation duration between participants of different skill levels were observed in the front, top, bottom, top right, and bottom right regions of the basket. Expert participants selectively stayed longer at each fixation point during the free-throw process, and the fixation points were stable, allowing them to quickly extract important information from a large amount of information and process it both quickly and accurately. Liang & Zhang (2012) found that lowering the number of fixations improves the efficiency of information search and processing. Extending the duration of each fixation point and using a relatively lower fixation rate can also improve the efficiency and accuracy of information processing. Time pressure also affects decision-making (Wang, 2009). Basketball rules stipulate that a free-throw shooter must shoot the ball within five seconds after the referee places the ball behind the free-throw line (Chinese Basketball, 2019). Athletes must complete target search positioning, information processing, action procedures, and psychological procedures within those five seconds. Standardized, long-term, systematic training gives these athletes the ability to optimize their fixation strategy and improve their ability to quickly search for targets and process information in the fixation area. It is impossible to search and analyze information in all areas within five seconds, so attention must be concentrated on the fixation positions that have a greatest impact on free-throw shooting accuracy. The expert group carried out fine information processing in the front, top, and top left regions of the basket during free throws, showing an efficient strategy for information processing at fixation points. This efficiency explains the differences in average fixation duration at different points among participants of different skill levels. Vickers (1996) explored the visual search characteristics of basketball players during free-throw preparation and also found that expert participants had longer fixation durations on the target area. Subsequent studies based on this finding determined that an increase in fixation duration improves free-throw shooting accuracy. Chuang, Huang & Tsung (2013) also reached this conclusion in their study of the attentional ability of basketball players using an electroencephalogram to record pre-free-throw shooting brainwave indicators. They found players who have relatively sustained focus before shooting have improved free-throw shooting accuracy. Zhang, Shen & Cha (2019) and Marten & Vickers (2004) also reached the same conclusion. Expert athletes have a higher percentage of fixation duration on critical areas of interest, indicating that more attentional resources are allocated to critical areas of interest, and the proportion of attention paid to non-critical areas of interest is significantly lower than that of novices.

Zhang, Shen & Cha (2019) found that the processing of visual information consumes limited attention resources, thus affecting key information processing, reducing processing efficiency. Novice basketball players lack specialized knowledge and have not received systematic training, resulting in multiple sources of interference during free-throw shooting, including technical movements and interference from the visual field. This leads to brief and unstable fixation durations, unclear attentional targets, and unstable fixation strategies and patterns resulting in a visual search process characterized by randomness and instability. Visual attention is driven by cognition, tasks, and goals (Zhang, Shen & Cha, 2019). The screening ability of basketball players when focusing on target information during free throws depends on their own mastery of basketball knowledge, specialized skill level, and accumulated experience. Expert players have clear attention goals during free throws and have formed a stable fixation pattern from long-term training and competition experience.

Pupil dilation

When processing information, the size of the pupil dilation changes, and the magnitude of the change is closely related to the mental effort expended during information processing. When the mental workload is high, there is a greater increase in pupil diameter. This indicator mainly reflects the degree of cognitive processing and cognitive load (Sui, Gao & Xiang, 2018).

The magnitude of the change in pupil dilation during free-throw shooting is closely related to the mental effort expended by the subjects during information processing. According to the experimental results, the maximum pupil dilation was found in the expert group, while the minimum was found in the novice group. Significant differences were found between the expert group and the novice and general groups, likely due to the level of attention concentration during free-throw shooting by the expert group. The expert group concentrate their attention on the target area during free-throw shooting with high levels of focus. Conversely, the novice and general groups lack the professional skills, theoretical knowledge, and experience needed to form a stable visual attention pattern during free-throw shooting. Therefore, their attention is more scattered and less focused, resulting in smaller pupil dilations compared to the expert group.

Heatmap

The fixation range of the expert group was more concentrated than that of the general and novice groups, with the novice group having the largest fixation range. The expert group had the longest fixation duration near the front of the basket, showing that expert players were selective in their visual fixation during free-throw shooting. The expert group demonstrated superior information search strategies and a higher degree of information processing compared to both the novice group and the general group. We propose that this disparity contributes to variations in free throw percentages among players of different skill levels. In the context of teaching or training, instructors or coaches should provide guidance to students or players, emphasizing the importance of information search and processing at the front of the basket when executing free throws. Through consistent practice, one can swiftly develop into an accomplished free throw specialist.

Conclusion

The expert group had a clear, stable visual attention pattern, with the focus of fixation concentrated on the front and top areas of the basket, and had less fixations and a longer fixation duration than the general and novice groups. The novice group showed dispersed and unstable visual attention and did not have a formed, stable, efficient fixation pattern. Profound basketball knowledge, specialized skills, and accumulated experience are the basis for efficient visual search and information processing during free-throw shooting, leading to high shooting percentages. Expert basketball players have an ability to make automatic visual calculations based on long-term training. Novice basketball players do not have the training or experience needed to form a stable visual attention pattern, leading to differences in gaze characteristics during free-throw shooting among athletes of different skill levels.

The attention process during free-throw shooting is not a static and stable process, but requires the quick and constant reallocation of mental resources to efficiently process visual information from the key areas of interest.

Supplemental Information

Supplemental Information 1 Data and figures.

Click here for additional data file.

Supplemental Information 2 Raw data.

Click here for additional data file.

Additional Information and Declarations

Competing Interests

Author Contributions

Human Ethics

Data Availability

The authors declare that they have no competing interests.

Chunzhou Zhao conceived and designed the experiments, performed the experiments, analyzed the data, prepared figures and/or tables, authored or reviewed drafts of the article, and approved the final draft.

Sunnan Li performed the experiments, analyzed the data, prepared figures and/or tables, authored or reviewed drafts of the article, and approved the final draft.

Xuetong Zhao performed the experiments, analyzed the data, prepared figures and/or tables, authored or reviewed drafts of the article, and approved the final draft.

The following information was supplied relating to ethical approvals (i.e., approving body and any reference numbers):

The Ethics Committee of School of Physical Education and Sports, Beijing Normal University approved the study (20221126).

The following information was supplied regarding data availability:

The datasets are available at Zenodo:

zhao chun zhou. (2023). Empirical study on Visual Attention Characteristics of basketball players of different levels during free-throw shooting original data. [Data set]. Zenodo. https://doi.org/10.5281/zenodo.7851565.

zhao chun zhou. (2023). “Empirical study on Visual Attention Characteristics of basketball players of different levels during free-throw shooting” AOI and Heatmap. Zenodo. https://doi.org/10.5281/zenodo.7851552.

Chun zhou Zhao. (2023). raw data [Data set]. Zenodo. https://doi.org/10.5281/zenodo.7962217.

Chun zhou Zhao. (2023). Raw data—Analysis of variance for fixation duration ,number of fixations ,pupil dilation ,saccadic amplitude of participants at different levels. [Data set]. Zenodo. https://doi.org/10.5281/zenodo.7962235.

zhao chun zhou. (2023). original data [Data set]. Zenodo. https://doi.org/10.5281/zenodo.7851393.

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
