# Peer review of "Empirical study on visual attention characteristics of basketball players of different levels during free-throw shooting"

_PeerJ, doi:10.7717/peerj.16607_

## Round 0.1 · original submission · Major Revisions

You need to carefully read the reviews and revise your manuscript.

Reviewer 1 ·

Basic reporting

- This study tried to adopt the eye-tracking technique in the context of a basketball free throw task. The idea of investigating gaze patterns in the real-world environment is something new to this line of research.

- It is an interesting question about where the shooters look during the free throw. However, the authors failed to develop a proper rationale for the study. In addition, DVs other than the points of gaze do not add any significant information to the study. Is knowing how the gaze are moving around an important issue?

Experimental design

- Line 37-43: There are reasons why previous experiments examined the gaze behaviors during re-active skills such as goalkeeping and baseball-hitting for the expert-novice comparison studies. Looking at various aspects of the opponent’s movements (or other environmental aspects) may provide us with important cues for upcoming events. In other words, athletes are trying to tease out a cue so they can better ANTICIPATE. In cases like free-throw or other closed skills, athletes need to focus visually on the target for successful execution. Yes, there are tons of studies examining the focus of attention (internal and external), but the idea of ‘fixating on a stationary target’ hasn’t been really challenged. In this case, we are talking about the ATTENTIONal focus. In the introduction, the authors mix up these ideas.

- Line 44-48: Not needed

- Line 58-70: Of course, aiming is a gaze behavior. According to the authors’ rationale, the most significant DV was supposed to be where the players aimed at (stable gaze position). However, the authors keep talking about other variables. Is visual search or visual extraction an important part of the attentional process during the free through? The authors are still confused about the difference between Gaze-Anticipation and Focus-Attention I described above.

- Line 136-183: Other than the Fixation duration variable, I cannot understand the rationales of reporting the DVs (especially pupil dilation) in this study.

- Also, the authors failed to describe how all the DVs were collected and calculated. Critical information regarding the time & events is missing. Especially, I have serious doubt that the authors fully understand what the saccadic amplitude represents.

Validity of the findings

- The main results were fixation duration, saccadic amplitude, and the number of fixations. All these variables are interrelated and confounded with movement time (or data collection period). But the manuscript came up with the discussion without introducing these issues.

Additional comments

- Throughout the manuscript: There are numerous ‘spacing’ mistakes between sentences. Must have happened during the pdf conversion. Please re-edit the whole manuscript. I started correcting these in the beginning but quit soon. Please refer to the memo on the pdf file.

- Throughout the manuscript: The statements suggested by authors are too often inaccurate. I can understand what the authors are implying, but they fail to deliver their intention accurately. Again, I started correcting these in the beginning but quit soon. Please refer to the memo on the pdf file.

Annotated reviews are not available for download in order to protect the identity of reviewers who chose to remain anonymous.

·

Basic reporting

This article discussed the differences in the characteristics of visual factors when shooting free throws through the detection of three different levels of basketball players examined by eye detectors. The authors took the eyes as the window of the individual soul, emphasized the importance of the brain's psychological process, and asked the subjects to wear eye tracker equipment and made free throws on the basketball court.

The focus position in the experimental parameters was divided into seven positions, which were: front of the basket, top right, top, top left, bottom right, bottom and bottom left. The research results found that the expert group was different from the general group and the novice group. and pupillary dilation were significantly different as well.

Basketball is one of the sports with high-intensity cardiorespiratory demands. It has emphasized as on-court competition in the line 44. The physical state of the subjects before free throw shooting was not clear, however, which might easily lead to misleading results, and which could be ambiguous with control physical fitness, ball skills and exercise intensity in terms of teaching and learning motor skills.

There are many errors in the format of punctuation marks in the full text, and the spaces before and after the punctuation are not clearly marked or left blank, which should be corrected through the function of editing software.

In addition, related text errors are as follows:

L16 top-right, and top-right…two areas or redundant?

L41 He et al(2016) modify needed to He et al. (2016)
conducive modify needed to conductive

L44 Please elaborate more about the crucial moment of visual attention associated with intense and close game situation in your experiment. The physical and mental status was not clear.

L70 As you might know, NBA players could make free throw with closed eyes. There were other more information irrelevant to visual information. Please explain your opinions.

L77 Please cite other expert-general-novice paradigm from other motor skill references. Explain the differences and mechanisms of the two or three paradigm.

L94 (Thieteen modify needed to Thirteen
How you defined the expert group in terms of A-level skill in sports. Please explain the similarities and differences between your expert level and international literature.

L95 I would suggest, Seven modify to 7 or seven

L98 Please elaborate the way to recruit your participants: general group from basketball specialty class and novice group from college students.

L150 In Table 2, p value is 0.13, not as written as p < 0.05. Please modify to statistical result.

L155-160 Please check and modify the description of fixation duration in terms of AOI.

L160-168 The description of LSD tests are lengthy and wordy. Please rewrite in terms of AOI for three groups.

L221 penalty kick technique? Is free throw used by legs? Please modify or explain it.

L282 missing alphabet, nd?

L282 spacing, changingnumber

L237 Please mark the literature and source of this statement.

L265 Please elaborate more on this sentence. Expert group participant’ visual search during penalty shooting is controlled top-down by cognition, while novice group participants are controlled bottom-up by cognition, and the general group falls somewhere in between. Most experimental results were not triplicate, but dichotomous.

L337, 341 repeated sentences. When the mental workload is high, the increase in pupil diameter is greater.

Experimental design

The procedures in the subject's consent form describe the way the subject raised the heartbeat. However, this part is not described in the experimental method. Please add this portion in your design.

Basketball is one of the sports with high-intensity cardiorespiratory demands. It has emphasized as on-court competition in the line 44. The physical state of the subjects before free throw shooting was not clear, however, which might easily lead to misleading results, and which could be ambiguous with control physical fitness, ball skills and exercise intensity in terms of teaching and learning motor skills.

L117 The experiment lasted 3 minutes and each participant made 3 free throws in a row. If the test fails or the procedure had something wrong, please explain whether there is any learning effect on focusing the rim.

Validity of the findings

This paper presented 1. No. of fixations, 2. Saccadic amplitude, 3. Fixation duration, 4. Pupil dilation and 5. Heatmap for results and findings. There were no the quantity or density data to run the statistical analyses of heatmap. In the heatmaps of all three groups revealed at the front of rim and more than one bright area. Please explain the heatmap associated with your experiment.

Table 3 It is suggested to add sum average and percentage column of AOI for three different groups.

L177-183 Heatmap seems to me that display the degree of attention of AOI. Please elaborate more on the difference of three groups. The heatmap displayed bright areas in the front site for all three groups in terms of concentration and scatter. Is there statistical analysis on the heatmap of three groups? The front area of AOI did not separate to three spots as top and bottom.

Additional comments

There are multiple ways to detect from the visual information in this paper. Gaze fixation is the main way for participant to obtain decision-making information. The number of fixations showed the information search strategy of subjects. The saccadic amplitude reflects the reading efficiency and material processing difficulty. Most importantly, the level of basketball players was depending on the level of processing detail and cognitive strategies. The paper examined the attention of target area during free throw shooting to explore three different levels of college female basketball players.

The author concluded that the gaze frequency of expert athletes is low and gaze duration is long. It is indicating the visual search efficiency of expert athletes is higher than that of general and that of novice athletes.

---

## Round 0.2 · Minor Revisions

The reviewer gives several suggestions. Please refer to it and revise it accordingly.

·

Basic reporting

The correspond author has answered my 11 comments one by one. They have asked English professionals to check for the language as well. For whose questions or comments what I raised, they have modified correctly in the paper too. In short, for comments of 1 to 6 and 11 are rational. Please work on the comments below.

For my seventh comment, I was expected that they would reply individually as merely those above contents have been revised.

For my eighth comment, the consent form about the intensity, they apologized that this experiment does not involve the exercise intensity of the participants. It has to be solved, otherwise, it might easily lead to mislead the results, and which could be ambiguous with control physical fitness, ball skills and exercise intensity in terms of teaching and learning motor skills.

For my ninth comment, please answer my question directly on whether three free throw caused any learning effect on focusing the rim during free throwing.

Experimental design

No comment.

Validity of the findings

Please work on the comments below.

For my tenth comment, please provide your statistical analysis for your AOI of heatmap.

Additional comments

I am looking forward to seeing the authors responses for my second review.

---

## Round 0.3 · accepted · Accept

You have completed all the revisions proposed by the reviewers. Therefore, I recommend your manuscript be published in PeerJ. Congratulations.